# Parallel Backpropagation for Shared-Feature Visualization

**Alexander Lappe**[1,2]    **Anna Bognár**[3]    **Ghazaleh Ghamkhari Nejad**[3]
**Albert Mukovskiy**[1]    **Lucas Martini**[1,2]    **Martin A. Giese**[1]    **Rufin Vogels**[3]
[1]Hertie Institute, University Clinics Tübingen    [2]IMPRS-IS    [3] KU Leuven
alexander.lappe@uni-tuebingen.de

## Abstract

High-level visual brain regions contain subareas in which neurons appear to respond more strongly to examples of a particular semantic category, like faces or bodies, rather than objects. However, recent work has shown that while this finding holds on average, some out-of-category stimuli also activate neurons in these regions. This may be due to visual features common among the preferred class also being present in other images. Here, we propose a deep-learning-based approach for visualizing these features. For each neuron, we identify relevant visual features driving its selectivity by modelling responses to images based on latent activations of a deep neural network. Given an out-of-category image which strongly activates the neuron, our method first identifies a reference image from the preferred category yielding a similar feature activation pattern. We then backpropagate latent activations of both images to the pixel level, while enhancing the identified shared dimensions and attenuating non-shared features. The procedure highlights image regions containing shared features driving responses of the model neuron. We apply the algorithm to novel recordings from body-selective regions in macaque IT cortex in order to understand why some images of objects excite these neurons. Visualizations reveal object parts which resemble parts of a macaque body, shedding light on neural preference of these objects.

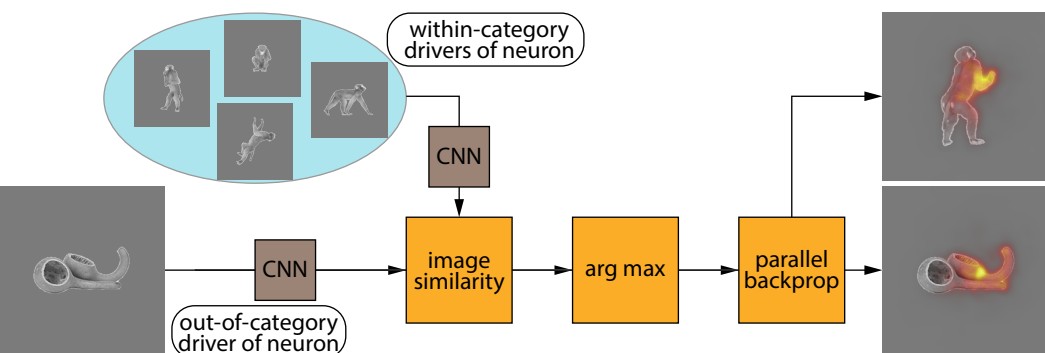

Figure 1: **Why does this object image activate IT neurons that are selective for bodies?** Our goal is to visually explain responses of category-selective neurons to outside-of-category (ooc.) stimuli. We start with an ooc. stimulus (object), that strongly activates a neuron which primarily fires for a specific category (bodies). We compute latent CNN activations for the image, as well as for a set of within-category reference images. A neuron-specific similarity metric operating on the latent activations finds the reference image most similar to the ooc. stimulus. The proposed parallel backpropagation method then highlights the shared features driving the neural response.

38th Conference on Neural Information Processing Systems (NeurIPS 2024).

# 1 Introduction

**Background.** The primate visual system has evolved to process a highly diverse set of tasks and stimuli. In higher visual areas, specialized subregions exist, in which neurons preferentially discharge in response to images stemming from a particular semantic class. These areas are often hypothesized to underlie specific computations that may only be relevant for a specific semantic concept, but it is not entirely clear why they emerge. The most prominent category-selective brain regions consist of 'face-cells' ('face patches'), which on average fire more strongly when stimulated with faces than objects [1, 2], as well as body-selective regions which have been studied to a lesser extent [3, 4, 5]. For face-selective cells, it has been shown that object images also elicit responses, albeit more sparsely than face images [6, 7]. Going further, [6] showed that models trained solely on non-face images could predict responses to face images in macaque face cells. Therefore, it has been argued that face-selective cells are not driven by the semantic concept of a face, but instead respond to visual features that are more common in face than object images. Hence, out-of-category (ooc.) images may still activate otherwise category-selective cells, as long as relevant features are apparent in the image. As of yet, characterization of these features has remained relatively rough, largely confined to the finding that face cells tend to prefer round objects and body cells tend to prefer spiky objects [8]. Other work suggests that only a small fraction of response variance in face cells can be explained by simple shape features like roundness [6]. While these global tuning properties are useful to understand average preferences of patches, we therefore argue that more fine-grained feature characterizations are necessary to understand responses at the single-image and single-neuron level.

**Contribution.** To address this gap, we propose a deep-neural-network-based method for visualizing features of an ooc. stimulus, which are responsible for eliciting high responses from an otherwise category-selective neuron. Our approach relies on analyzing the similarity of the image to a selected within-category image which displays similar features, as shown in Fig. 1. Specifically, after finding a within-category image with similar, neuron-specific features, we use gradient methods to highlight features extracted from a vision model which are shared between the two images and drive the neural response. This helps to visually answer the question why a feature detector that is preferably activated by within-class images, would also respond to the ooc. image in question. The procedure is compatible with any backbone visualization method that returns a saliency map for hidden units of a convolutional neural network. Further, it is entirely class-agnostic, and can therefore be used for any selective brain region, as well as for studying internal behaviour of artificial vision systems. In this work, we show results for a novel set of multi-unit recordings from body-selective regions in macaque superior temporal sulcus. Body cells constitute a particularly interesting problem, as different body poses produce vast variability among body images, suggesting a rich set of features driving these neurons. We summarize our contributions as follows:

- We present a novel method for visualizing features driving responses of category-selective neurons to out-of-category images, shedding light on why these neurons do not exclusively respond to within-category images,

- We present results from multi-unit recordings of body-selective neurons in macaque IT cortex, demonstrating that these neurons encode overlapping visual features for bodies and objects,

- We apply the proposed visualization method to the data, discussing why some non-body objects activate these neurons.

# 2 Related work

**Category-selective visual brain areas.** The largest part of the body of work studying category selectivity in the brain is targeted towards face patches [2, 1]. Several papers have come to the conclusion that face-selective neurons respond to visual features correlated with faces, rather than the semantic concept of a face. [7] generated maximally exciting images for face cells, which activated the neurons strongly but were not rated as face-like by human participants. [9] showed that face cells also respond selectively to pareidolia images, which are objects eliciting perception of a face in humans. Neurons still responded after the images were scrambled, destroying perception of a face, indicating that responses were driven by low-level features. Recently, [6] successfully predicted responses of face cells to face images, after training a linear readout model using solely non-face

images. [8] proposed a unified view of IT cortex organization, arguing that selectivity was based on a small set of principal axes of image space. Reducing the dimension allows the authors to state that face cells prefer objects with high scores on principal components corresponding to 'stubby' and 'animate', whereas body-selective cells respond to 'spiky' and 'animate' objects. Recent work [10] challenges the hypothesis of shared coding principles between faces and non-faces in face cells, showing that computational mechanisms in these brain regions are far from being well understood. Our neurophysiological recordings add additional evidence to this debate. Highly relevant to our work is that of [11], which visualized body cell responses to bodies and some objects by showing fragments of a highly activating image to determine most important image regions. This approach showed that a large proportion of cells respond to local body fragments. The main advantages of our computational approach are its high-throughput, allowing to visualize a large number of cells and objects simultaneously, and enhancing interpretability of features by analyzing them in the context of the preferred semantic class.

**Attribution methods.** A substantial body of work exists on attributing behaviour of deep computer vision models to specific image regions. The most common goal is to understand why classification models output the observed class label, given an input image, meaning that attribution methods are often applied to the very last network layer. Several families of approaches have been proposed for tackling this problem: The first relies on image perturbations like occlusion [12, 13], the second is based on an analysis of latent activations [14, 15], and the third computes gradients of class activations w.r.t. pixel intensities [16, 17, 18]. More recent work [19, 20] has shown that some of these methods rely too strongly on the input image itself, being independent of the network to varying degrees. Even though our reweighting scheme is compatible with any attribution method that is computable for latent activations, we therefore rely on vanilla backpropagation in this work, which has been shown to not be biased towards reconstructing the input image. We provide results for integrated gradients [18] in the appendix. Further, relevant to our work is that of [21] which computes saliency maps for image similarity. The approach is similar to that of Grad-CAM, as it relies on analyzing feature activations before and after a global pooling layer at the end of the network. For that reason, the latter method is mainly suitable for visualizing global similarity as judged by a network trained to do so, rather than more local features relevant for biological neurons along the visual hierarchy in the primate brain.

## 3 Methods

We propose an approach for visual explanations of neural responses to out-of-category (ooc.) stimuli in category-selective visual brain areas. The categories currently of most interest in the high-level vision literature are faces and bodies, but the method can be applied to any semantic category. We leverage differentiable neuron models to explain high neural firing rates in response to an ooc. image by analysing visual similarity to an image from the preferred category which also drives the neuron. Formally, let $x_{\text{out}} \in \mathbb{R}^{3 \times h \times w}$ denote an ooc. image yielding high activity from a recorded category-selective neuron. The overall goal of this work is to determine the visual features of $x_{\text{out}}$ driving the neural response. We approach this problem in three steps:

1. We learn a linear readout vector $w$ on top of a pretrained CNN $f(\cdot)$ to predict neural responses to within-category stimuli $x_{\text{in},1}, \ldots, x_{\text{in},N}$. The learned vector then incorporates information about which features apparent in within-category images are relevant for the neural response.

2. We employ a neuron-specific similarity metric based on the learned readout to find a within-category image $x_{\text{in}}$ with similar visual features as $x_{\text{out}}$. Visual inspection of this reference image on its own can yield insights on why $x_{\text{out}}$ activates the neuron.

3. We backpropagate gradients of CNN activations to the pixels of both images and reweight them to highlight features that are

   (a) present in $x_{\text{out}}$,
   (b) present in $x_{\text{in}}$,
   (c) highly relevant for the neural response.

   Explicitly highlighting shared features helps identify specific image regions responsible for the images' neuron-specific similarity.

### 3.1 Modelling neural responses

In recent years, a pretrained CNN backbone combined with a trainable linear readout module has become the gold standard in modelling the stimulus-driven variance in neural responses to images [22, 23]. In these models, an image $x \in \mathbb{R}^{3 \times h \times w}$ is fed through a CNN $f(\cdot)$ up to a predetermined layer to obtain a latent representation $a \in \mathbb{R}^c$. The predicted response for a neuron is then computed as

$$\hat{y} = \langle a, w \rangle, \tag{1}$$

where $w \in \mathbb{R}^c$ is a learned weight vector, and $\langle \cdot, \cdot \rangle$ denotes the standard dot product. The vector $w$ encapsulates information about which visual dimensions the neuron is tuned to, as a large $w^{(i)}$ implies that increased activity in feature $i$ will strongly increase the predicted neural response. Conversely, $w^{(i)} = 0$ implies no change in predicted activity if feature $i$ is manipulated and thus such features can be ignored when trying to understand a neuron's response pattern (assuming a perfect model fit). Therefore, training a model to predict neural responses to within-category stimuli reveals which visual features drive responses within the category. If the model then generalizes to ooc. stimuli, we can infer that ooc. neural responses are driven by features present in both data distributions.

### 3.2 Neuron-specific image similarity

If an image $x_{\text{out}}$ strongly drives a neuron, we propose to interpret the relevant image features by studying a visually *similar* image from the category that the neuron is selective to. To quantify similarity, we adapt the common method of computing the cosine similarity of latent activations computed using a CNN [24]. Since we have additional information on which latent dimensions drive the neuron, we weight feature $i$ by the corresponding weight $w^{(i)}$. Formally, for images $x_1$, $x_2$ and their corresponding latent activations $a_1 := f(x_1)$, $a_2 := f(x_2)$, we define the neuron-specific image similarity as

$$s(x_1, x_2) := \frac{\langle a_1 \odot w, a_2 \odot w \rangle}{||a_1 \odot w||_2 ||a_2 \odot w||_2}, \tag{2}$$

where $\odot$ denotes the Hadamard product. Since $w$ is often sparse, this formulation will effectively ignore most dimensions and focus solely on those that are highly relevant for predicting the neural response. For the downstream procedure of generating a visual explanation for the neural response to the image $x_{\text{out}}$, we simply select the most similar image from our within-category dataset, i.e.

$$x_{\text{in}} = \arg\max\{s(x_{\text{out}}, x) : x \in D_c\}, \tag{3}$$

where $D_c$ denotes the set of within-category images.

### 3.3 Parallel backpropagation for shared-feature visualization

Even though the images $x_{\text{out}}$ and $x_{\text{in}}$ ideally have a high similarity metric, the similar features are not always clearly identifiable. In some cases, the features might not be easily interpretable, or in other cases, several features might be shared upon visual inspection, such that it remains unclear whether all of them or a subset contribute to neural activity.

In order to further highlight shared features contributing to model activity, we compute weighted gradients of the latent model activations w.r.t. to image pixels [16]. This method was originally devised for classification models to highlight those image pixels that most strongly influence a given class probability. We build on this work to visualize shared features between images by introducing a simple reweighting procedure. Specifically, we strengthen the influence of features that are shared, while attenuating features that are specific to only one of the images. For the sake of brevity, we illustrate the procedure for the gradients of $x_{\text{out}}$ only, as the process is analogous for $x_{\text{in}}$. First of all, note that the gradient of the predicted neural response $\hat{y}_{\text{out}}$ is given by

$$\frac{\partial \hat{y}_{\text{out}}}{\partial x_{\text{out}}} = \frac{\partial}{\partial x_{\text{out}}} \langle a_{\text{out}}, w \rangle = \sum_i w^{(i)} \frac{\partial}{\partial x_{\text{out}}} a_{\text{out}}^{(i)}. \tag{4}$$

Due to the sum rule of calculus, the gradient of the predicted response is given by a weighted sum of the gradients of the latent features, with $w^{(i)}$ acting as weight for feature $i$. Hence, features that are highly relevant for model predictions will dominate the gradient. In turn, those pixels for which an

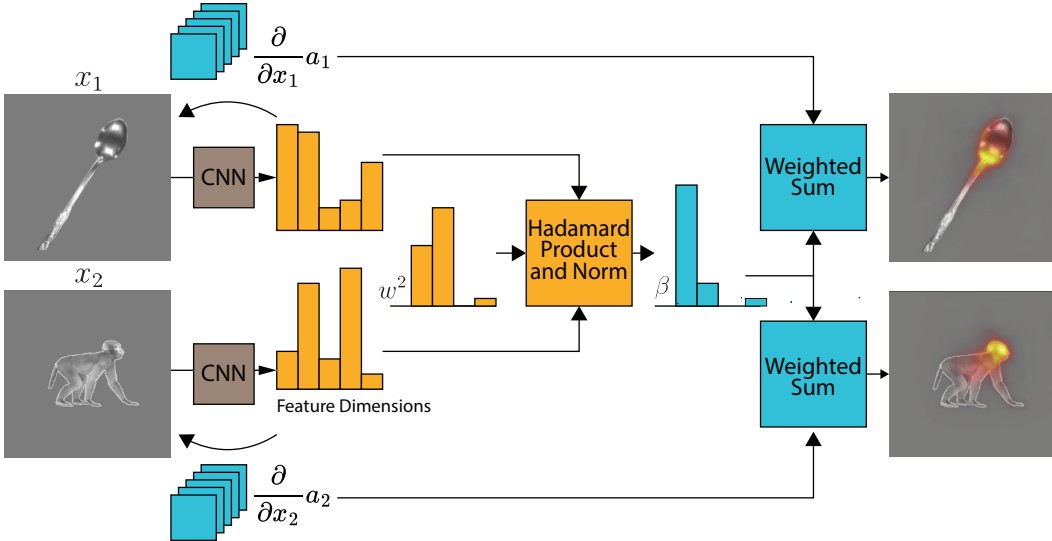

Figure 2: Sketch of the parallel backpropagation method. A pre-trained CNN cut off at a predetermined layer computes latent feature activations. These are backpropagated to obtain the Jacobians of the two activation vectors w.r.t. to the respective images. We then calculate the normalized Hadamard product of the activation vectors and the element-wise square of the learned linear readout vector for the considered neuron. The pixel saliency map is then computed as the sum of gradients of each feature, weighted by the feature's entry in the Hadamard product.

infinitesimal increase would strongly enhance one of the highly relevant features will be assigned a high intensity when plotting the gradient over the image.

For our purposes, this gradient map is not satisfactory. First of all, it does not take the other image into account at all, and thus fails to leverage the images' similarity structure. Further, since the gradient weights do not carry information on whether a feature is present in the image in the first place, it is theoretically possible to assign high intensity to pixels that would strongly increase features with low activity. To remedy these deficiencies, we propose an adjusted pixel saliency map $I$ based on replacing the weights for the features in (4).

Before reweighting the features, we first smooth and normalize each gradient to have unit norm, i.e. $||\frac{\partial}{\partial x_{\text{out}}} a_{\text{out}}^{(i)}||_2 = 1$. Smoothing alleviates pixel-level noise and allows the user to determine *regions* of pixels with high contribution. It has been shown that a post-processing smoothing step substantially improves the quality of gradient-based attributions [20]. Normalization ensures that the pixel saliency map is not determined solely by the gradient magnitude, as some features may be more sensitive to pixel perturbation than others. More importantly, it allows us to later bound the norm of the saliency map from above, which substantially improves interpretability. Subsequently, we weight each feature by its contribution to the neuron-specific similarity metric $s(\cdot, \cdot)$ given in (2). Formally, we define the weight for feature $i$ as

$$\beta^{(i)} := \frac{a_{\text{in}}^{(i)} w^{(i)} a_{\text{out}}^{(i)} w^{(i)}}{||a_{\text{in}} \odot w||_2 ||a_{\text{out}} \odot w||_2}, \tag{5}$$

and finally the pixel saliency maps

$$I(x_{\text{out}}; x_{\text{in}}, w) := \sum_i \beta^{(i)} \frac{\partial}{\partial x_{\text{out}}} a_{\text{out}}^{(i)}, \qquad I(x_{\text{in}}; x_{\text{out}}, w) := \sum_i \beta^{(i)} \frac{\partial}{\partial x_{\text{in}}} a_{\text{in}}^{(i)}. \tag{6}$$

The revised weight vector $\beta$ imposes high weights only on those features that are highly activated in the feature vectors of both images (captured by $a_1^{(i)}$ and $a_2^{(i)}$), and are also relevant for the neuron's activity (captured by $(w^{(i)})^2$). Dividing by the product of norms allows us to bound the intensity of

the saliency maps by writing

$$||I(x_{\text{out}}; x_{\text{in}}, w)||_2 = ||\sum_i \beta^{(i)} \frac{\partial}{\partial x_{\text{out}}} a_{\text{out}}^{(i)}||_2 \leq \sum_i ||\beta^{(i)} \frac{\partial}{\partial x_{\text{out}}} a_{\text{out}}^{(i)}||_2$$

$$= \sum_i \beta^{(i)} \underbrace{||\frac{\partial}{\partial x_{\text{out}}} a_{\text{out}}^{(i)}||_2}_{=1} = \sum_i \beta^{(i)} = s(x_{\text{in}}, x_{\text{out}}). \tag{7}$$

Note that this inequality only holds if $a_{\text{in}}^{(i)} \geq 0$ and $a_{\text{out}}^{(i)} \geq 0$ for all $i = 1, \ldots, c$, as this is needed to ensure $\beta^{(i)} > 0$. However, this constraint is usually satisfied as latent activation are commonly fed through a ReLU layer before extraction. The inequality yields that the total intensity of the pixel saliency map (for either image) as given by its $L_2$ norm is bounded by the neuron-specific similarity. Thus if the images are dissimilar, the saliency maps will have low intensity. Finally, note here that the reweighting procedure is agnostic to the way the salience map for each feature is generated. Any (backpropagation) pixel-attribution method may be used, as long as each latent feature is assigned one saliency map with unity norm.

Running these steps successively for one recorded neuron yields as output one ooc. image and one within-category image, along with one saliency map per image. To study shared features, we display the images side-by-side with the saliency maps overlayed, as seen in Figs 4 and 5.

## 4  Experimental setup

**Model architecture.**   For experiments, we use a Resnet-50 [25], which was adversarially trained on ImageNet [26, 27], as our CNN backbone. This architecture has shown good fits to neural data in several studies [28, 29, 30]. To predict the response to an image, we feed it through the Resnet up to the last ReLU of layer 4.1. Subsequently, a Gaussian readout [31, 28] reduces the dimension of the Resnet activations from $c \times h_a \times w_a$ to $c$ by selecting a learned spatial location in the latent feature map from which to extract the final features. The receptive fields of the extracted features cover the entire foreground of the images. Finally, a fully-connected layer maps these features to neural responses.

**Stimuli.**   For gathering neural data, we utilized two sets of images. The first consisted of 475 images of a macaque avatar on a gray background (see Fig. 1, 2). The images were subsampled from a set of 720 images, in which the avatar appeared in 45 unique poses, extracted from nine different behavioural classes, each shown from 16 viewpoints [32]. The second set comprised 6,857 objects from varying categories shown on the same gray background and was combined from the OpenImages dataset [33], as well as several smaller ones [34, 35]. To test for body selectivity, we used an additional set of 2068 control body images including a variety of species. These images were only used to test for category-selectivity of the recorded cells. All other references to body images in the text refer to the original image set showing the monkey avatar. All stimuli were centered with respect to the fixation point. They were shown to the monkey at a resolution of 280x280, and were resized to 224 for the Resnet.

**Neural data collection.**   We recorded multi-unit activity (MUA) from and surrounding two fMRI-defined body category-selective patches [5] in the macaque superior temporal sulcus (STS), using 16-channel Plexon V probes, while the subjects performed a fixation task. Since body patch neurons typically discharge sparsely for non-body objects, we employed online stimulus selection: In a first phase, we recorded responses to the set of 475 monkey avatar images. We then trained a model for each channel to predict neural responses to these images. Subsequently, we predicted neural responses to our set of object images, and for each neuron we selected the highest and lowest predicted activator, as well as the object most similar to the top-activating body image, according to $s(\cdot, \cdot)$. We followed the same procedure for the control body images. Finally, in a second experimental phase we recorded responses to these novel images as well as a subset of 75 of the original body images, to test recording stability. We included recording channels in the analysis if the test/retest stability was higher than .60, as measured by the correlation between responses to body images before and after model fitting. Further, we tested each channel for body-category-selectivity by comparing the median response to the selected objects and the selected control bodies using a Mann-Whitney U-test. Both the animal

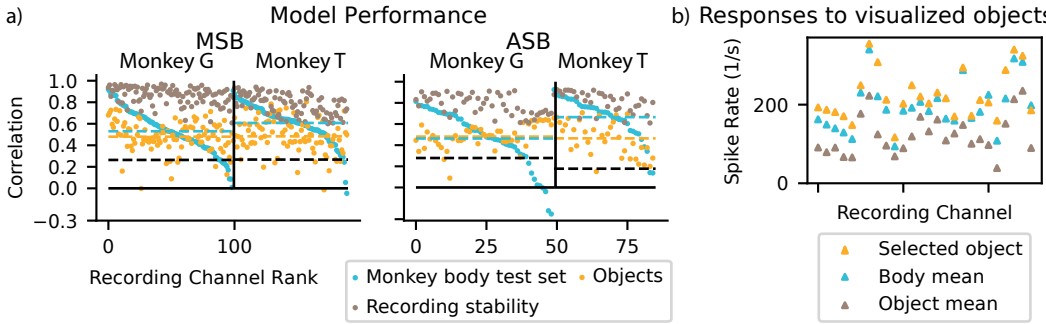

Figure 3: a) Correlation between predicted and recorded neural responses for held-out monkey body images (blue) and object images (orange). Significant positive correlation for object images demonstrates that the visual features predictive of responses to body images are also predictive of responses to objects. Brown dots show the correlation between responses to the same stimuli in the first and second recording phase. Dashed lines show the average across channels (colored) and the .05 confidence interval for the correlation coefficient under the null hypothesis that $\rho = 0$ (black). b) Neural responses to the objects for which features are visualized in Figs. 4 and 5. Responses to visualized objects are higher than mean responses to objects and bodies in the vast majority of cases.

care and experimental procedures adhere to regional (Flanders) and European guidelines and have been approved by the Animal Ethical Committee of Leuven under the protocol number P182/2019. Further experimental details are given in the appendix.

**Fitting the neuron model.** We split the monkey image set into a training/validation/test split consisting of 400/50/25 images. The parameters of the Gaussian readout location, as well as the linear weights, were trained simultaneously using the Adam optimizer [36], to minimize the mean squared error between recorded and predicted responses. We augmented the training data by incorporating silhouettes of the monkey, for which the tuning of body patch cells in mid-STS has been shown to be largely preserved [37]. Further, we penalized the readout-weights using $L_{1/2}$ regression to sparsify the vector while still allowing for some large weights. We set the learning rate to $10^{-4}$ and the weight of the regularization to 0.1 After training for 2500 epochs, we selected the model with lowest loss on the validation set. Importantly, the model was not trained to predict responses to non-body images, meaning that it must utilize the same visual features to predict bodies and objects. Models and training runs, as well as the visualization procedure were implemented in PyTorch [38]. To compute the Jacobian of latent features for visualization, we utilized the corresponding functionality of PyTorch's autograd. All experiments were run on a single Nvidia RTX 2080Ti.

## 5 Results

### 5.1 Model generalizes from bodies to objects

After training the models on body images, we first test how well they generalize to images of objects. Model performance in terms of correlation between predicted and recorded neural response is shown in Fig. 3 a). 93.7/95.3 % of channels in the posterior/anterior region exhibit a significant positive correlation, suggesting an at least partially class-agnostic feature preference. Interestingly, strong performance on the held-out monkey images does not seem to be a necessary condition for strong performance on objects. This effect may be caused by body images being highly similar due to fine-grained sampling of poses and viewpoints, compared to high feature variance among objects.

### 5.2 Feature visualization

**Shared features between objects and bodies.** Having found that there exists a set of visual features driving neural responses to both objects and bodies, we apply the proposed visualization method to characterize these features. Results are displayed in Figs. 4 (posterior region MSB) and 5 (anterior region ASB). We show results from multi-unit sites for which model performance on the out-of-category images was high ($r > 0.4$), selecting a subset representing a wide range of

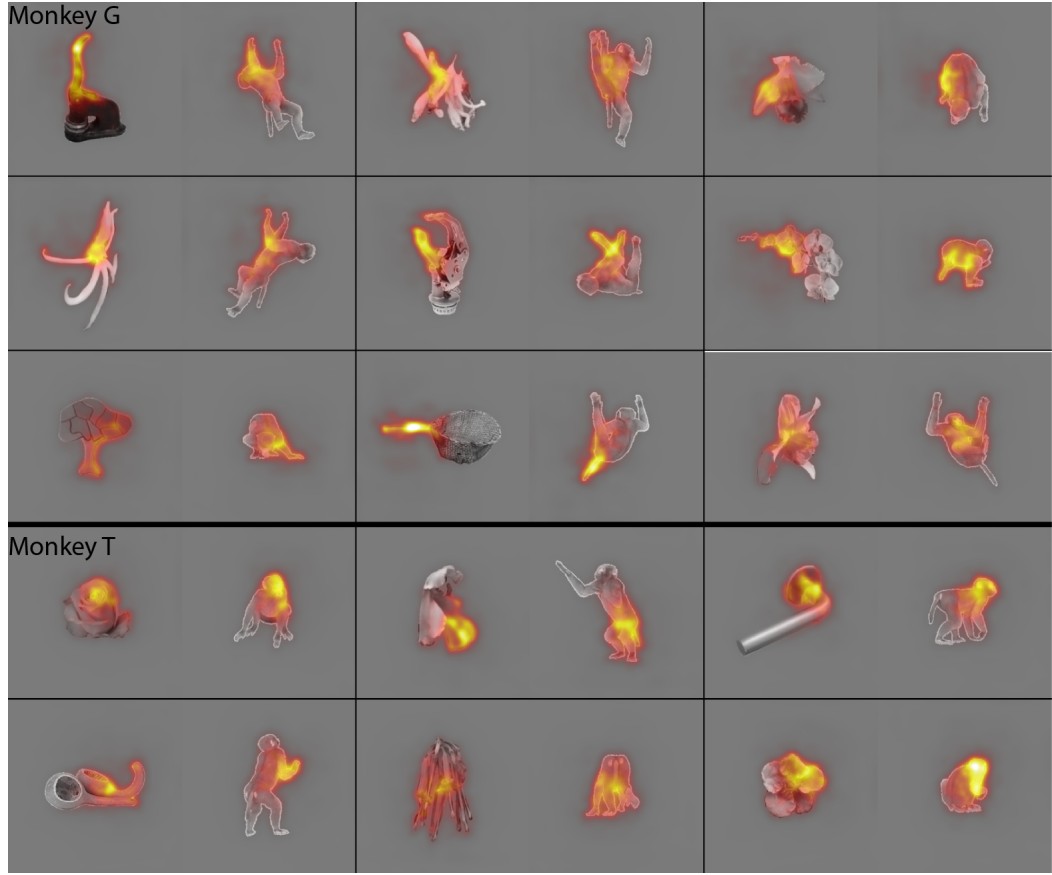

Figure 4: Results obtained by applying the proposed method to multi-unit recordings from body-selective cells in macaque STS (posterior region). Each subplot corresponds to one recording channel. The object on the left is among the most highly activating objects for the channel. The image on the right corresponds to the most similar preferred body.

highlighted features. For neighbouring recording sites, visualizations are often similar, likely due to similar feature preferences. For each channel, we display the image pair with highest neuron-specific similarity among the top-5 activating objects and top-15 activating bodies. Fig. 3 b) demonstrates that the visualized objects activate the corresponding channels more strongly than the average body/object image in most cases.

The method discovers a variety of shared features between highly activating bodies and objects. Most of them correspond to parts of the body rather than the entire body, which is aligned with previous findings for neurons from MSB.[11]. In fact, specific object parts seems to bear resemblance to specific body parts in the model's latent space. For example, extended structures appear to activate the same latent dimensions as arms/shoulders, so a model that relies on arms/shoulders to explain responses to body images also predicts strong responses to other extended structures. We observe objects driving the model due to similarity to limbs, tails, heads, torsos as well as more diffuse features which are more difficult to interpret. Interestingly, while a a lot of the observed features could be characterized as 'spiky', we also find neurons which are activated by stubby objects. The corresponding bodies show crouched poses without protruding extremities. This demonstrates that at the single-image and multi-unit level, tuning properties are more fine-grained than previously suggested [8]. In some cases, we observe the same object with different highlighted features in different recording channels, indicating that objects may have multiple features that activate different neurons. An example of this can be seen in Fig. 5, where the highlighted features of the same image correspond to a leg and a torso of the effective body images of two different channels. Additional results are given in the appendix.

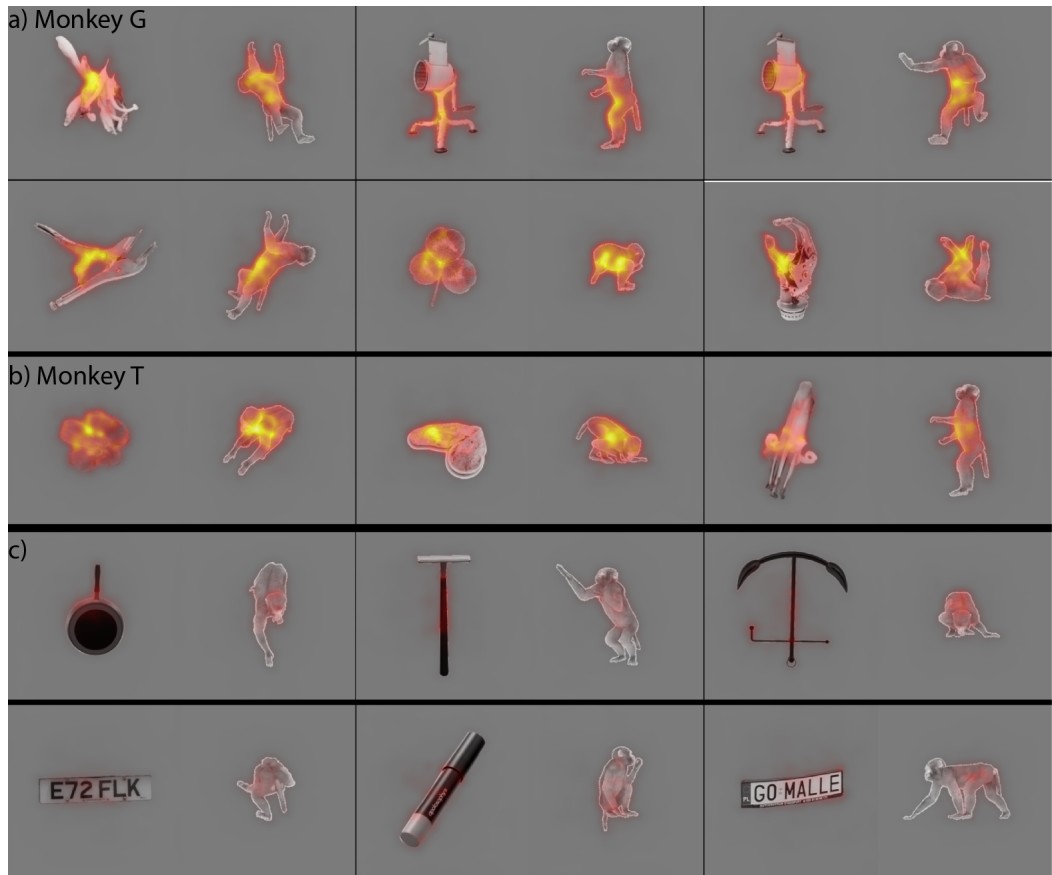

Figure 5: a)/b) Results for ASB region. c) Example results for objects lacking a highly similar body image among the channel's top drivers, and for weakly activating objects. Top row: strongly driving objects. Bottom row: weakly driving objects. Low similarity is properly reflected by low intensity of the saliency map.

**Objects without shared features.**   We find that for some object images eliciting a high neural response, the visualization method yields no shared features with any of the top-activating body images (Fig. 5 c)). This could be either due to neurons preferring features not present in the original body image set, or due to tuning properties not captured by the model. These cases demonstrate the utility of the bound for the norm of the saliency maps given in (7), as the lack of image similarity is clearly reflected by the visualization.

## 6   Discussion

**Limitations.**   The proposed approach is made possible through the use of a differentiable neuron model, which means that the visualization quality depends on the ability of the model to capture neural tuning properties. As models of visual processing further improve in the future, we predict that visualization quality will improve accordingly. Further, the visualizations will reflect idiosyncrasies of the underlying attribution method used for generating saliency maps for latent features. Since this backbone can be chosen freely, advances on the topic of attribution methods will also improve our visualizations.

**Conclusion.**   We presented a method for visualizing selectivity of class-selective visual feature detectors when confronted with out-of-class images. Further, we showed that body-selective neurons encode bodies and objects using an at least partially shared feature set. We visualized these features, providing an explanation for why some objects activate body-selective neurons. Future work could involve using the same method for other category-selective areas, like face patches. Additionally, one

could test these visualizations by presenting highlighted fragments to the subject in a closed-loop fashion.

## Acknowledgments and Disclosure of Funding

The authors thank Rajani Raman and Prerana Kumar for valuable discussions about this project. Further, the authors thank C. Fransen, I. Puttemans, A. Hermans, W. Depuydt, C. Ulens, S. Verstraeten, S. T. Riyahi, J. Helin, and M. De Paep for technical and administrative support. AL, AB, GKN, AM, LM, MG and RV are supported by ERC-SyG 856495. MG is supported by HFSP RGP0036/2016, BMBF FKZ 01GQ1704. The authors thank the International Max Planck Research School for Intelligent Systems (IMPRS-IS) for supporting Alexander Lappe and Lucas Martini. Parts of the stimulus images are due to courtesy of Michael J. Tarr, Carnegie Mellon University, http://www.tarrlab.org/.

**Contributions.** AL, AB, MG and RV conceptualized the study. AL developed and implemented the visualization method. AB, GGN and RV collected the neural data. AL and AB analyzed the data. AM, AB, LM and AL contributed to stimulus generation. AL wrote the initial draft of the manuscript, and all authors contributed to the final version. MG and RV supervised the project.

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

# A Appendix / supplemental material

## A.1 Additional Results

### A.1.1 Synthetic data

To test our method on a wider set of semantic categories, we generate six synthetic, category-selective neurons. To do so, we gather a small set of within-category images $x_{\text{in},1}, \ldots, x_{\text{in},N}$ and a set of out-of-category images $x_{\text{out},1}, \ldots, x_{\text{out},M}$ from the stimulus sets used for the electrophysiological experiments. Feeding these through the CNN and sampling a spatial location using a Gaussian readout with random location preference yields activation matrices $A_{\text{in}} \in \mathbb{R}^{N \times c}$ and $A_{\text{out}} \in \mathbb{R}^{M \times c}$. A synthetic neuron with readout weights $w \in \mathbb{R}^c$, that on average prefers the within-category images can then easily be found by solving

$$\arg\max_w (A_{\text{in}}w)^\mathsf{T} A_{\text{in}}w - (A_{\text{out}}w)^\mathsf{T} A_{\text{out}}w = \arg\max_w w^\mathsf{T}(A_{\text{in}}^\mathsf{T} A_{\text{in}} - A_{\text{out}}^\mathsf{T} A_{\text{out}})w.$$

This formulation is useful since the Rayleigh quotient is solved by setting $w$ to be the first eigenvalue of $A_{\text{in}}^\mathsf{T} A_{\text{in}} - A_{\text{out}}^\mathsf{T} A_{\text{out}}$ [39]. Of course, the Resnet also contains category-selective neurons that do not need to be constructed artificially. However, we aim to make the experiment as similar to neural recordings as possible, where model neurons are usually given as a linear readout of latent activations.

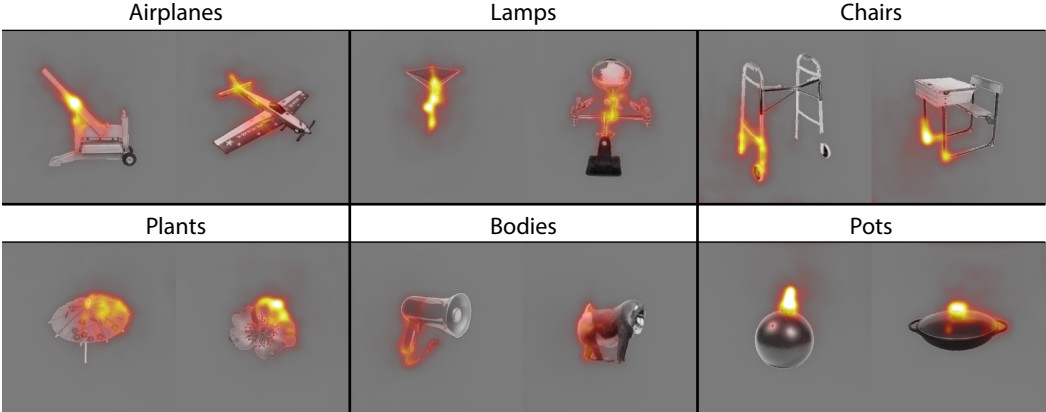

Figure 6: Results of applying the parallel backpropagation method to six synthetic, category-selective neurons. Each row corresponds to one neuron, with the preferred category indicated above.

We observe that the visualizations clearly mark features that are common among within-class images, showing why the ooc. stimuli drive the otherwise category-selective neurons.

### A.1.2 Integrated Gradients

Our visualization procedure is based on a weighted sum of a tensor containing saliency maps for all latent features. As discussed in the main text, the method is agnostic towards how the individual saliency maps are generated. For the main experiments, we use the vanilla gradient method which results in reweighting the Jacobian of the latent features. Here, we experiment with using a different saliency backbone, namely Integrated Gradients [18]. For an image $x$, Integrated Gradients approximates the path integral of the gradients along the straightline path from an image of zeros to $x$. Originally developed for visualizing scalar outputs, we adapt the formula from [18] to the multi-dimensional case to yield

$$\text{IntegratedGrads}(x) = x \odot \frac{1}{m} \sum_{k=1}^{m} Jf\left(\frac{k}{m}x\right),$$

where $Jf(x)$ denotes the Jacobian of the CNN features w.r.t. $x$. Since [19] found that this formulation is heavily influenced by the element-wise multiplication with the input $x$, we omit this step in the computation. Results shown in 7 demonstrate that the visualizations are almost indistinguishable from those computed using vanilla backpropagation.

## A.2 Additional details for neural data collection

**Experiment details.** We recorded neuronal responses during passive fixation task using a 2x2 degree fixation window (EyeLink 1000 infrared eye tracker sampling at 1000 Hz). Stimuli were shown gamma-corrected on a

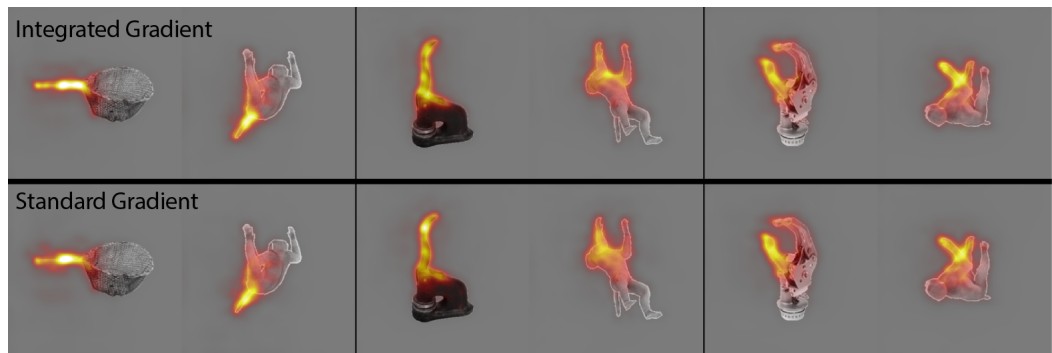

Figure 7: Results for computing the Jacobian using the Integrated Gradient method (top row). Bottom row shows the results for the standard gradient computation from the main text for reference.

22.5-inch ViewPixx monitor at a distance of about 57 cm, with a resolution of 1920x1080 and a refresh rate of 120 Hz. A set of 475 stimuli of a monkey avatar in various poses were presented 8 times in a pseudorandom sequence for 200 ms on a gray background, with a 250 ms interstimulus interval. During the interstimulus interval, only a fixation dot was present and monkeys could receive a brief juice reward . Stimulus onset and offset were indicated by a photodiode, detecting luminance changes synced with the stimuli, in the display corner (invisible for the monkeys). Control of stimulus presentation, event timing, and juice delivery was managed by an in-house Digital Signal Processing-based computer system, which also monitored the photodiode signal and tracked eye positions. Neuronal data was collected from and surrounding two fMRI-defined body-selective patches in the ventral superior temporal sulcus (STS) using 16-site linear electrodes (Plexon V probe) with Open Ephys acquisition board and software (sampling rate: 30000 Hz, filtered between 500-5000 Hz). Multi-unit activities were extracted using Plexon Offline Sorter, after applying a high-pass Butterworth filter with a cutoff frequency of 250 Hz. From the stimulus event synchronized continuous data, 550 ms trials were extracted, comprising 200 ms prestimulus and 150 ms poststimulus periods. Responsive (at least for one stimulus showing stronger than 5 spikes/s net responses (baseline: -75 to +25 ms, response: +50 to +250 ms)), and body-selective MUAs (p<0.01 Kruskal-Wallis test), with a split-half reliability >0.5 (Spearman Brown corrected) were selected. The responses of these MUA sites were used to predict neural responses to our set of object and body images, and for each neuron we selected the highest and lowest predicted activator, as well as the object most similar to the top-activating avatar stimuli, according to $s(\cdot, \cdot)$. Finally, in a second experimental phase we recorded responses to these object and body images as well as a subset of the original monkey avatar stimuli, to test recording stability (same experimental design as in the first phase). For all neurons considered in this work, we tested for body-selectivity by comparing the median response to bodies to the median response to objects using a Mann-Whitney U-test, considering only channels for which the test detected a significant difference.

**Animals and husbandry.** Two male 7 years old rhesus monkeys (Macaca mulatta), weighing 9.2 and 11 kg, respectively, contributed to this study. The animals were housed in enclosures at the KU Leuven Medical School and experienced a natural day-night cycle. Each monkey shared its enclosure with at least one other cage companion. On weekdays, dry food was provided ad libitum, and the monkeys obtained water, or other fluids, during experiments until they were satiated. During weekends, the animals received water along with a mixture of fruits and vegetables. The animals had continuous access to toys and other forms of enrichment. After fMRI scanning, we implanted a custom-made plastic recording chamber, allowing a dorsal approach to temporal body patches. In each animal, the location of the recording chamber was guided by the fMRI body localizer described in [40]. Surgery was performed using standard aseptic procedures and under full anesthesia.

