# OpenReview forum: "Parallel Backpropagation for Shared-Feature Visualization"
_NeurIPS.cc/2024/Conference — NeurIPS 2024 spotlight_

### Official Review · Reviewer_aP8u · 2024-07-10

**Soundness:** 3
**Presentation:** 4
**Contribution:** 3
**Rating:** 6
**Confidence:** 4

**Summary:**

This paper presents an innovative method for explaining why an IT neuron that is supposed to tune to a particular object class or category would respond to outside-of-category stimuli. The proposed method uses parallel backpropagation to highlight the features in the out-of-category stimuli that are shared by the stimuli in the tuned class. The method helps discover some shared features between highly activating bodies and objects in IT neurons but also fails in other cases.

**Strengths:**

The method is original, innovative, sound, and very clearly presented. It can be useful for IT neurophysiological research to resolve some mysteries in the neural codes in IT.

**Weaknesses:**

The paper focuses on the presentation of the method. The presentation of the scientific results is not very systematic. They are shown as anecdotal illustrations of some success and failure cases. Hence, the scientific insights provided are a bit preliminary, and the significance of the results is perhaps limited. The contribution is solid but perhaps a bit thin.

**Questions:**

Perhaps the usefulness of the method is more general than the authors envisioned.  Can the method be applied to neurons in V4 and V1?  Can it be used to analyze artificial deep networks to understand "neural codes" in CNN?

Also, the assumption of the approach is that neurons are tuned to one single category, but could neurons be tuned to multiple categories or multiple features?

**Limitations:**

There are no concerns about the limitation issue.

---

> ### Author Rebuttal · Authors · 2024-08-02
>
> Thank you for the valuable comments and the encouraging feedback. Regardings questions/criticisms:
> > The presentation of the scientific results is not very systematic [... and] a bit preliminary.
>
> We agree that the results on feature selectivity in macaque body patches can be expanded on to get a more complete picture of their tuning properties for both bodies and non-body objects. As you stated, we put the focus of the paper more towards a rigorous introduction of the visualization method. While this approach has its drawbacks, we argue that it also has advantages.
> Ideally, we would like to see the proposed method applied to not only body patches, but other category-selective regions as well to shed light on the general concept of category-selective processing in the brain. To make that feasible, we aimed to introduce the procedure in a detailed fashion to facilitate easy use by other researchers. Further, focusing on the class-agnostic parts of the paper may make it interesting for the broader audience at NeurIPS. We would also like to point out that this is, to the best of our knowledge, the first work on visualizing the features underlying category selectivity in IT cortex.
> That being said, we agree that a more systematic approach would be of interest and future work will focus on that.
> > Can the method be applied to neurons in V4 and V1?
>
> This is an intriguing question, since semantic-category selectivity is mostly studied in IT. One interesting approach could be to replace the semantic selectivity by selectivity for a specific shape/texture descriptor. As a naive example, a neuron that responds to semicircles could be classified as a (seemingly) circle-selective neuron, as the semicircle feature is more common in the image distribution of circle images than in the global distribution. Applying the proposed method should then highlight that the neuron responds to semicircles because they are a subfeature of the more general shape descriptor. While this example is somewhat obvious, application of our method might be of use when considering the complex feature tunings in V4.
> > Can it be used to analyze artificial deep networks?
>
> We would argue that the method can indeed be used for studying neural codes in CNNs. One interesting case is an application to a classification model's readout layer.  Here, category-selectivity is vitally important since any response to out-of-category images is interpreted as evidence for an incorrect hypothesis. Consider an image $x$ which is incorrectly classified as category $c$.  One could  then use a pool of images from category $c$, find the most similar image to $x$ according to $s(\\cdot,\\cdot)$ and visualize the shared features. This could yield insight into why $x$ is misclassified, and why the model uses those specific features as evidence for class $c$. One could also study category selectivity in hidden layers of vision models. We provide a proof-of-concept for this case in the appendix (A.1.1), where we compute visualizations for artificially created category-selective units.
>
> > Could neurons be tuned to multiple categories?
>
> The question of a neuron being tuned to multiple categories is an interesting one, as definitions of a category can be quite fuzzy (consider, for example the case of face neurons and body neurons ). However, let’s assume that a neuron is tuned to two semantically rather different categories. In that case, our work, together with previous papers, would suggest that it is because there are shared features that are common in the image distributions of both categories. In that case, the proposed method could still be used to visualize these shared features. One could do this by sampling images from the two categories and feeding pairs of them through the visualization pipeline.
> > Could neurons be tuned to multiple features?
>
> We identify two avenues in which a neuron's possible tuning for multiple different features relates to the proposed method. The first is the possibility that a neuron likes different features across categories, i.e. feature A for category 1 and feature B for category 2, where feature A and B are the results of different computational processes.
>
> We employ two approaches to ensure that the method does not highlight these as shared features: First, we train the neurons' readout model only on within-category images and then test on outside-of-category images. Good performance on outside-of-category images implies that the features driving the responses are (at least partially) shared between the two categories. Second, the visualization method is able to detect cases in which features are not shared between two images. Considering the similarity metric in equation (2), the activation vectors $a_1$ and $a_2$ will be dominated by different dimensions if the features between the two images are not shared, yielding a small dot product and thus a small similarity $s$. Since the norm of the attribution map is bounded by the image similarity $s$  (equation (7)), the attribution map will have low intensity, reflecting that the two images drive the neuron due to different features.
>
> The second relevant avenue is the situation in which a neuron is activated by multiple within-class features (e.g. arms and torsos for bodies). If the ooc. stimulus to be visualized contains only one of the features, this should be reflected by the method. The weighting for feature $i$  in the attribution map, as given in equation (5), is $a_{\\text{in}}^{(i)} a_{\\text{out}}^{(i)}(w^{(i)})^2$, yielding that feature $i$ will be ignored if it is absent in either of the images, even if the neuron is tuned to it. Conversely, if the neuron is tuned to multiple features and these features are apparent in both images, they will all be highlighted.
> Therefore, we argue that the proposed method is able to deal with cases in which a neuron is tuned to a variety of different features.

---

> > ### Comment · Reviewer_aP8u · 2024-08-13
> > **Responses to Authors' rebuttal**
> >
> > I would like to thank authors' interesting discussion on the questions I raised.  I do think this is a valuable contribution.

---

### Official Review · Reviewer_5Zdi · 2024-07-11

**Soundness:** 4
**Presentation:** 4
**Contribution:** 4
**Rating:** 10
**Confidence:** 3

**Summary:**

A deep-learning-based approach is proposed in the paper "Parallel Backpropagation for Shared-Feature Visualization" to visualize shared visual features in high-level visual brain regions, which are typically thought to respond selectively to particular categories like bodies or faces. Despite this selectivity, these neurons occasionally respond to stimuli outside of their category, possibly due to shared visual characteristics. By backpropagating activations to the pixel level and enhancing shared features while attenuating non-shared ones, the authors present a strategy for highlighting these shared features using a deep neural network to model neural responses and identify relevant features.

**Strengths:**

The text is very well structured, and the questions and methodology are written in a very clear and detailed manner. This work follows an innovative approach which presents a novel and exciting way to use DL in order to gain insights of single neuron selectivity across the visual hierarchy. The method's steps—which include parallel backpropagation, determining neuron-specific picture similarity, and training a linear readout on top of a CNN—are fully explained. Finally, this work includes an original experimental component that differentiates from a lot of other applications of DL on already existing datasets.
A strategy for visualizing shared elements that drive brain reactions to out-of-category stimuli is presented in this research, which shows promise. The method has the potential to further our understanding of how the brain processes visual information, even though there is still room for improvement.

**Weaknesses:**

In this work, the method is heavily dependent on the deep neural network used, and how effectively it can fit the neural data. Fortunately, the cases where the tunning properties are not captured by the model can be identified as mentioned in the end of the results section. The ability to generalize to other categories such as face patches was not tested and remains to be tested.
The technique is predicated on the idea that within-category and out-of-category photos share a significant amount of attributes. It is not clear what we expect for the model to do and how useful outcomes are going to be in situations where there is little overlap.
Finally, while the single neuron tuning specificity is the main topic of this work, it remains to be seen how a similar approach but with population representations compares.

**Questions:**

Subsequent research ought to concentrate on enhancing the underlying models and attribution methods as well as verifying the approach in various scenarios such as with different categories and in a closed-loop manner. The authors already acknowledge these directions.

**Limitations:**

The authors have a limitation section that addresses their work's assumptions.

---

> ### Author Rebuttal · Authors · 2024-08-02
>
> Thank you for your review and comments. We highly appreciate your confidence in our work. We agree that the mentioned limitations are largely adressed in the paper, but for the sake of completeness we post some remarks here.
> > The ability to generalize to other categories such as face patches was not tested and remains to be tested.
>
> We agree that this is a highly interesting next step and are planning to apply this method to study tuning properties of both body and face patches in more depth in future work.
> > The technique is predicated on the idea that within-category and out-of-category photos share a significant amount of attributes. It is not clear what we expect for the model to do and how useful outcomes are going to be in situations where there is little overlap.
>
> If a neuron is driven by a feature set $A$ for category-stimuli and a disjoint feature set $B$ for out-of-category stimuli, we agree that the proposed method would be unable to generate meaningful explanations, since the tuning for features $B$ simply cannot be explained (solely) by studying the tuning for features $A$. However, in that case the CNN activation vectors $a_{\\text{in}}$ and $a_{\\text{out}}$ should have distinct activation patterns, leading to a low image similarity $s(\\cdot,\\cdot)$ (equation (2)) which then implies an attribution map with low intensity (equation (7)). This allows the user to detect such cases. Furthermore, while we certainly don't claim that such neurons don't exist, the majority of neurons we recorded coded for similar features across categories (subsection 5.1).
> > Finally, while the single neuron tuning specificity is the main topic of this work, it remains to be seen how a similar approach but with population representations compares.
>
> This would indeed be an interesting comparison. At the population level, we hypothesize that most if not all features of the within-category image activate the population due to variations in tuning properties of single neurons. For an out-of-category stimulus, the method should then highlight the subset of features present in that image as since only those features will be weighted according to equation (5).

---

> > ### Comment · Reviewer_5Zdi · 2024-08-09
> >
> > I would like to thank the authors for the response to my comments.

---

### Official Review · Reviewer_7NES · 2024-07-17

**Soundness:** 4
**Presentation:** 4
**Contribution:** 4
**Rating:** 8
**Confidence:** 4

**Summary:**

The paper shows that a hypothesis for the brain (category-selective neurons are actually selective to generic lower-level features that are present in those categories, not necessarily features specific to those categories) can be reproduced and visualized in a ResNet trained to predict neural activity. If I understand correctly, it is capable of generating hypothetical pairs of images with similar features (as in A.1.1). One could then test if an actual neuron responds similarly to the pair as the model predicts. Or find a real neuron, record, then based on responses, use the model to get a stream of stimuli that should elicit similar responses (because of similar low-level features) and show it to the real neuron to see if it responds as we expect.

**Strengths:**

This is a clear and straightforward method that contributes to tools available for analyzing what drives individual units in regression-to-neural-data deep learning models.

Comprehensive and clear background/related work; very clear writing overall and description of methods on high and low levels. I got an intuition pretty quickly for what the method was even before the equations.

**Weaknesses:**

The results are interesting and intriguing to look at (Figure 4), however at the end of the day it comes across as a bit anecdotal as opposed to giving us insight into general principles of shape or object representation.  This is not a strong criticism though - I accept it is a good starting point, and still a worthwhile contribution.

Also, I note that all the objects are presented on blank backgrounds.  I wonder how things would change when the objects or bodies are presented in their natural context within a visual scene?

other comments:
111: Maybe this is standard terminology I'm not familiar with, but "readout vector" confused me at first because it makes it sound like it's logits or something? But it's just the weights from the latent representation to a single output unit. Line 131 more clearly calls it "learned weight vector"
145: should be a_2=f(x_2)?

It would be interesting to see more mathematical/geometric analysis of the shared features (beyond stubby vs spiky) in the future, e.g. joints at a specific angle, combinations of contours in a certain way, etc.

**Questions:**

see above

**Limitations:**

see above

---

> ### Author Rebuttal · Authors · 2024-08-06
>
> Thank you for reviewing the manuscript, and for your positive feedback.
>
> > The results are interesting and intriguing to look at (Figure 4), however at the end of the day it comes across as a bit anecdotal as opposed to giving us insight into general principles of shape or object representation. This is not a strong criticism though - I accept it is a good starting point, and still a worthwhile contribution.
>
> We understand this criticism, as we steered the focus of the manuscript towards introducing the method in a clear and class-agnostic fashion to allow others to easily apply it to their own recordings of category-selective neurons. We acknowledge that further work should attempt to get a more systematic overview of the shared coding properties.
>
> > Also, I note that all the objects are presented on blank backgrounds. I wonder how things would change when the objects or bodies are presented in their natural context within a visual scene?
>
> We decided for neutral backgrounds to allow for good control of low-level image features.
> Further, we wanted the model to pick up on as few spurious image statistics as possible to enable it to generalize across categories. Since it seems like similarity between highly activating bodies and objects is primarily driven by local features / parts, we would now hypothesize that the tuning properties would be be preserved when presenting stimuli as parts of naturalistic scenes. We agree that it would be worthwhile to test this hypothesis.
>
> >  Maybe this is standard terminology I'm not familiar with, but "readout vector" confused me at first because it makes it sound like it's logits or something? But it's just the weights from the latent representation to a single output unit. Line 131 more clearly calls it "learned weight vector" 145: should be a_2=f(x_2)?
>
> Thank you for pointing out the unclear language - we have adjusted these parts in the text.
>
> > It would be interesting to see more mathematical/geometric analysis of the shared features (beyond stubby vs spiky) in the future, e.g. joints at a specific angle, combinations of contours in a certain way, etc.
>
> We agree that it would be of high interest to find quantitative feature descriptors to go along with qualitative visualizations. We are concurrently working on further ways of characterizing the features driving these neurons.

---

> > ### Comment · Reviewer_7NES · 2024-08-14
> > **Thanks for your response.**
> >
> > Good paper, I have nothing further to add.

---

### Official Review · Reviewer_yQac · 2024-07-17

**Soundness:** 3
**Presentation:** 3
**Contribution:** 2
**Rating:** 5
**Confidence:** 4

**Summary:**

The authors proposed a deep learning based method to visualize shared features in neurons that are selective to specific categories, such as faces or bodies, when they respond to out-of-category stimuli. The method identifies visual features driving the selectivity of neurons by modeling responses to images based on latent activations of a deep neural network. The paper highlights the application of this method to body-selective regions in the macaque IT cortex, demonstrating that neurons encode overlapping visual features for bodies and objects. This approach provides insights into why certain non-body objects activate body-selective neurons and offers a more fine-grained understanding of neural responses.

**Strengths:**

The paper is well-structured, with a clear abstract, introduction, methodology, and results sections.

While the proposed method is based on well-established deep learning techniques, specifically leveraging latent activations of a deep neural network to model neuron responses, the application to novel recordings from body-selective regions in macaque IT cortex demonstrates its practical utility and empirical soundness.

I think the primary contribution lies in providing a tool that allows for a more fine-grained investigation of neuron responses in visual neuroscience. By revealing why certain non-body objects activate body-selective neurons, the paper contributes valuable insights to the understanding of neural selectivity and visual processing.

**Weaknesses:**

The paper primarily applies existing deep learning techniques rather than introducing new machine learning algorithms or models. This might be seen as a limitation for those expecting significant advancements in machine learning methods, especially readers from NeurIPS.

**Questions:**

Can The authors clarify the novel contributions of the proposed method in comparison to existing visualization techniques? How does the approach provide unique insights that are not achievable with current methods?

How generalizable is the method to other types of neurons or different brain regions beyond body-selective areas in the macaque IT cortex? Can the proposed approach be adapted to study other semantic categories or species?

What are the broader implications of these findings for the field of visual neuroscience? We knew from previous studies that out-of-category stimuli can also activate neurons coding for in-category features.

**Limitations:**

The authors have addressed the limitations.

---

> ### Author Rebuttal · Authors · 2024-08-06
>
> Thank you for review and comments. We hope to be able to increase your confidence in recommending the manuscript for publication.
>
> > The paper primarily applies existing deep learning techniques rather than introducing new machine learning algorithms or models. This might be seen as a limitation for those expecting significant advancements in machine learning methods, especially readers from NeurIPS.
>
> It is correct that we do not introduce new models or learning algorithms. Our contribution is more in line with efforts towards explainability, as we propose a novel method to study the behaviour of a trained model with fixed weights. We do believe that explainability methods can be of interest to a relatively broad community of people working on vision. Further, we argue that applications to neuroscience are of special importance to the community at NeurIPS, in order to foster a relationship between the two related fields.
>
> > Can The authors clarify the novel contributions of the proposed method in comparison to existing visualization techniques?
>
> Our method can be understood as an approach to turn any (single-image) attribution method into a (multi-image) similarity-attribution method by reweighting feature-wise attribution maps. In doing so, it is completely agnostic to the underlying visualization technique used to compute these feature-wise attributions. Previous work on similarity attribution methods (see related work) focuses on output layers to visualize global image similarity, while we are interested in local features that drive responses of neurons in the brain. Therefore, our approach is applicable to (functions of) hidden units.
>
> > How does the approach provide unique insights that are not achievable with current methods?
>
> We are interested in why an out-of-category (ooc.) stimulus $x_{\\text{out}}$ activates an otherwise category-selective neuron.
> One could use an existing visualization technique to study the neuron's preference of $x_{\\text{out}}$. This would provide an attribution map over the image, highlighting the visual features driving the model neuron's response. However, it does not yield insight into *why* the image would activate a category-selective neuron, specifically. We argue here, in line with previous work, that neural preference of these ooc. images is due to shared features with the preferred category. Therefore, a proper visual explanation should also include how the driving features in $x_{\\text{out}}$ occur among images of the preferred class, to understand why preference of these features has developed. The proposed method therefore explains neural responses to an ooc. image by providing a strongly driving category image, highlighting the preferred features in the category image, and then highlighting the corresponding features in the ooc. stimulus.
>
> > How generalizable is the method to other types of neurons or different brain regions beyond body-selective areas in the macaque IT cortex? Can the proposed approach be adapted to study other semantic categories or species?
>
> Yes, the method is completely category-agnostic and therefore can be used for any semantically selective brain region, as well as selective units in artificial networks. Future work could apply the method to, e.g., face cells as another category in more depth; as a preliminary example, consider Fig. 4 panel (4,1) where we visualize an object driving a face preferring-neuron. For details regarding category agnosticism, we kindly refer to section 3 of the manuscript.
>
> > What are the broader implications of these findings for the field of visual neuroscience? We knew from previous studies that out-of-category stimuli can also activate neurons coding for in-category features.
>
> Regarding broad implications, we first provide additional evidence for the hypothesis of category-agnostic tuning properties in macaque IT cortex. There is still conflicting evidence regarding this question [1], necessitating further work.
>
> Furthermore, given the finding that out-of-category stimuli can activate category-selective neurons, the interesting question becomes why this is the case. Our work supports the notion that such stimuli exhibit features which are also common in images of instances from the preferred category. This is, to the best of our knowledge, the first attempt at visually characterizing these features. Our findings suggest, that the objects activating body-selective neurons have local parts that are visually similar to local body parts. We hypothesize that this finding transfers to other category-selective areas. For example, [2] found that simple shape descriptors like 'roundness' are not enough to explain face cells' responses to objects. In our experiments, we also found cells that fire in response to the head; their best-driving objects are not round, but our method finds parts which resemble a head (Fig. 4 (4,1), (4, 2), (5,3)). This yields insight into *why* these objects might activate the neurons, beyond descriptors like 'roundness'
>
> [1] Shi Y, Bi D, Hesse JK, Lanfranchi FF, Chen S, Tsao DY. Rapid, concerted switching of the neural code in inferotemporal cortex. bioRxiv [Preprint].
>
> [2] Kasper Vinken et al. ,The neural code for “face cells” is not face-specific.Sci. Adv.9

---

> > ### Comment · Reviewer_yQac · 2024-08-12
> > **Thanks for the response**
> >
> > I went through the answers from the authors and am happy with them. However, I am still concerning about the techenical novelty after going through the paper again. I will keep my score.

---

### Author Rebuttal · Authors · 2024-08-06

We would like to thank all reviewers for their time and their valuable insights. We were glad to see that all reviewers found the presentation of the work clear and understandable, which is also reflected by the fact that all summaries clearly capture the main points of the paper. We also did not identify any factual errors in the reviews. Further, we appreciate that all reviewers suggest that the manuscript should be accepted for publication, albeit with varying confidence. We hope that we were now able to clarify remaining questions, and properly addressed the raised issues. Even though there was partial overlap between comments, we answered all of them individually to make the rebuttals for each reviewer self-contained. If some of our answers were not to the point and you would like to discuss further, we are happy to do so over the next couple of weeks.

---

### Decision · Program_Chairs · 2024-09-25

**Decision:**

Accept (spotlight)

**Comment:**

This paper describes a novel approach for identifying visual features that are common between a preferred object class vs. other object classes with an application to characterizing the neural selectivity of visual neurons.

The reviewers noted that while the deep learning techniques are relatively well established, their application to visual neuroscience remains relatively novel and compelling. On the negative side, reviewers commented that the reported results and analyses remain limited and appear to be relatively preliminary.

All in all, there appears to be strong support from the reviewers to accept this paper, which introduces a novel, broadly applicable method for better understanding neural selectivity in visual areas. Hence, the AC recommends that the paper be accepted.